# Computed Tomographic Changes in Patients with Cystic Fibrosis Treated by Combination Therapy with Lumacaftor and Ivacaftor

**DOI:** 10.3390/jcm10091999

**Published:** 2021-05-07

**Authors:** François Arnaud, Nathalie Stremler-Le Bel, Martine Reynaud-Gaubert, Julien Mancini, Jean-Yves Gaubert, Guillaume Gorincour

**Affiliations:** 1Service d’Imagerie Médicale, AP-HM Hôpital Nord, 13015 Marseille, France; 2Centre de Ressource et de Compétences de la Mucoviscidose (CRCM) Pédiatrique, AP-HM Hôpital la Timone, 13005 Marseille, France; nathalie.stremler@ap-hm.fr; 3Centre de Ressources et de Compétences de la Mucoviscidose (CRCM) Adulte, AP-HM Hôpital Nord, 13015 Marseille, France; martinelouise.reynaud@ap-hm.fr; 4Département de Santé Publique, Aix-Marseille Université, APHM, INSERM, IRD, SESSTIM, Hôpital de la Timone, BIOSTIC, 13005 Marseille, France; julien.mancini@ap-hm.fr; 5Service d’Imagerie Médicale, AP-HM Hôpital de la Timone, 13005 Marseille, France; jeanyves.gaubert@ap-hm.fr; 6Institut Méditerranéen d’Imagerie Médicale Appliquée à la Gynécologie, la Grossesse et l’Enfance (IMAGE2), 6 rue Rocca, 13008 Marseille, France; guillaume.gorincour@gmail.com

**Keywords:** computed tomography, cystic fibrosis, lumacaftor, ivacaftor, CFTR regulator

## Abstract

Background: As Cystic Fibrosis (CF) treatments drastically improved in recent years, tools to assess their efficiency need to be properly evaluated, especially cross-sectional imaging techniques. High-resolution computed tomography (HRCT) scan response to combined lumacaftor- ivacaftor therapy (Orkambi^®^) in patients with homozygous for F508del CFTR has not yet been assessed. Methods: We conducted a retrospective observational study in two French reference centers in CF in Marseille hospitals, including teenagers (>12 years old) and adults (>18 years) who had received lumacaftor–ivacaftor and for whom we had at disposal at least two CT scans, one at before therapy and one at least six months after therapy start. CT scoring was performed by using the modified version of the Brody score. Results: 34 patients have been included. The mean age was 26 years (12–56 years). There was a significant decrease in the total CT score (65.5 to 60.3, *p* = 0.049) and mucous plugging subscore (12.3 to 8.7, *p* = 0.009). Peribronchial wall thickening (PWT) was significantly improved only in the adult group (29.1 to 27.0, *p* = 0.04). Improvements in total score, peribronchial thickening, and mucous pluggings were significantly correlated with improvement in FEV1 (forced expiratory volume in 1 s). Conclusions: Treatment with lumacaftor–ivacaftor was associated with a significant improvement in the total CT score, which was mainly related to an improvement in mucous pluggings.

## 1. Introduction

Cystic fibrosis is an autosomal recessive genetic disease caused by mutation in the gene that codes for the cystic fibrosis transmembrane conductance regulator (CFTR) protein, regulating the transport of sodium and chlorine ions through epithelial cells [1]. These abnormalities are responsible for an alteration in the function of many organs, dominated by the impairment of respiratory function.

In the past 50 years, and despite the lack of specific treatment available until recently, average life expectancy has increased considerably, from five years in the 1960s to about 40 years now [1]. F508del, the most common CFTR mutation that causes CF, is found in up to 80% to 90% of people with CF with more than 50 percent of these patients being homozygous for this mutation.

Lumacaftor–ivacaftor therapy is the first specific treatment for patients with homozygous F508del mutation. The efficacy and safety of this dual therapy has been demonstrated in phase 3 clinical trials. The drug acts as a corrector and potentiator for the CFTR protein (CFTR modulator). It combines lumacaftor (which is designed to fix the defective CFTR protein, and can move to the proper place on the cell surface) with ivacaftor (which helps improve the function of the protein as a chloride channel on the cell surface) [2,3]. 

The United States Food and Drug Administration and the European Medicines Agency approved the use of this combined therapy in patients homozygous for the F508del mutation in 2015. The French National Authority for Health did the same in 2016. However, the heterogeneity of the clinical response, leading some teams to define predictive factors to the efficacy, and the cost of treatment have hampered its use in some countries [4,5,6].

Recently, a study conducted by Burgel et al. in the 47 CF reference centers in France has evaluated the efficacy and safety of this lumacaftor–ivacaftor combined therapy in a real-life post-approval setting, on 845 CF patients (292 adolescent >12 years, 553 adults) from 1 January to 31 December 2016. This real-life study confirmed the efficacy of such treatment with improvement in lung disease (ppFEV1), nutritional status body mass index (BMI) and decrease in number of intravenous antibiotic courses in patients who tolerated combined therapy [7]. However, evolution computed tomographic (CT) changes, and assessment of bronchial and parenchymal lesions under therapy has not been specifically evaluated. With increasingly reduced irradiation doses, chest CT offers a sensitive and accurate alternative to spirometry in the therapeutic follow-up of these CF patients [8].

The aim of our study was to assess HRCT changes in cystic fibrosis patients treated with lumacaftor–ivacaftor. The secondary objectives were to look for a correlation between CT scan changes, ppFEV1, and sweat chloride test. 

## 2. Materials and Methods

### 2.1. Study Population

A retrospective observational, non-randomized, and non-blinded study has been conducted in two French cystic fibrosis centers from Marseille University Hospitals, (Hôpital de la Timone and Hôpital Nord, Assistance Publique-Hôpitaux de Marseille, France) carried out from January 2016 to June 2019. Adolescents (>12 years) and adults (>18 years) homozygous for F508del mutation who received combined therapy with lumacaftor and ivacaftor were included in this study. All subjects underwent a baseline CT scan prior to initiation of treatment and a follow-up CT scan after at least six months of treatment. The treatment consisted of two daily oral doses (200 mg lumacaftor/125 mg ivacaftor per tablet). Patients and their parents (for minors) were informed that the tablets had to be taken with a fatty food. Patients were included only if the treatment had been well conducted until the follow-up CT scan. All patients or parents approved the study and signed an informed consent.

### 2.2. Clinical Data Collection

The clinical data included age, gender, weight, height, body mass index (BMI), and forced expiratory volume in one second (FEV_1_). Sweat chloride was obtained only in the pediatric (adolescent) population (12 to 18 years old). 

### 2.3. HRCT Imaging

We analyzed two CT scans, one performed prior to treatment initiation (baseline CT scan) and a follow-up CT scan performed after at least six months of well-conducted treatment. CT scans were performed during routine follow-up, in clinically stable patients. The adolescent population was evaluated on a Siemens Somatom Definition scanner (Timone Hospital, Siemens, Erlangen, Germany) and the adult population was evaluated at North Hospital, on a GE Optima 660 CT scanner (General Electrics, Milwaukee, Wisconsin). The acquisition parameters varied from 80 to 120 kilovolts (kV) with systematic use of an automatic dose modulation system. Two acquisitions were performed, without injection of contrast medium, one in inspiration and the other in expiration, with the use of a spatial reconstruction filter. The Dose-Length Product (DLP) was collected for each scanner. CF structural lung disease was evaluated by one radiologist using the Brody Scoring system who was trained by a thoracic radiologist. Each scanner was evaluated using the modified Brody scoring method. The choice was made not to multiply the readings because the intra- and inter-observer reproducibility of the Brody score had already been evaluated [9,10]. CT scans were not anonymized, and the radiologist could compare baseline CT scan and follow-up CT scan.

### 2.4. Modified Brody Scoring Method

Each lung was divided into three lobes (culmen and lingula separated on the left), and for each lobe the presence, extension, and severity of the lesions classically induced by cystic fibrosis were evaluated: mucous plugging, peribronchial wall thickening, bronchiectasis, parenchymal opacities, and air trapping (Appendix A). Bronchiectasis was identified by a bronchoarterial ratio >1, non-tapering bronchus, a bronchus within 1 cm of the costal pleura, or a bronchus abutting the mediastinal pleura. Peribronchial wall thickening was defined as a bronchial wall thickness >2 mm in the hila, 1 mm in the central lung and 0.5 mm in the peripheral lung. Mucous plugging was defined as an opacity filling a defined bronchus or presence of either dilated mucous-filled bronchi or peripheral thin branching structures or centilobular nodules. Air trapping was defined as areas of lung on the expiratory images that remained similar in attenuation to the appearance on inspiratory. The parenchyma score depended on the existence of condensations, ground glass opacities, cysts, or bullaes. Extension defined the percentage of involvement in the central and peripheral lung (absence, less than one third, between one and two thirds, and more than two thirds of a lobe).

All the subscores obtained were added together and then normalized on a scale of 100 to obtain a total score.

### 2.5. Statistical Analysis

Continuous variables are presented as mean ± (minimum–maximum) or mean ± standard deviation. Their pre- and post-evolutions were tested using paired-t tests or paired Wilcoxon tests (when *n* < 30). The Pearson correlation coefficient was used to study the relationship between the different evolutions. The threshold of significance for all bilateral tests was set at *p* < 0.05. Statistical analyses were performed using IBM SPSS Statistics 20.0 (IBM Inc., Armonk, NY, USA).

## 3. Results

Thirty-four patients treated with lumacaftor–ivacaftor were included. In the study cohort there were 21 adolescents (62%) and 13 adults (38%). The sex ratio was balanced (17 males and 17 females). The mean at inclusion age was 26 years old (12–58 years old). The time between the performance of CT scans and the beginning of treatment was quite variable. The average time from baseline CT scan to initiation of treatment was 6.2 months (0–36 months). The average time from initiation of treatment and follow-up CT scan was 15.4 months (7–54 months). The average time between the two scans was 21.4 months (7–61 months). Radiation doses (DLP—Dose Length Product) averaged 230.7 mGy.cm for children (95.0–458.6) and 226.3 mGy.cm for adults (123.4–1044.0). Expired acquisition was not performed in 10 patients. There was no sweat chloride testing in 13 adults.

The main results are summarized in Table 1.

At follow-up, there was a statistically significant decrease in Brody’s total score. There was also a significant decrease in the mucous plugging subscore (Figure 1). The peribronchial thickening and hyperinflation scores were non-significantly improved. There was no significant change in the subscore of bronchiectasis and lung parenchyma abnormalities. In addition, there was a significant improvement in BMI and sweat test. FEV1 was not significantly improved.

The Pearson correlation test found a significant association between improvement in FEV1 and improvement in total score (r = −0.51, *p* = 0.002), peribronchial thickening (r = −0.43, *p* = 0.01), and mucous plugging (r = −0.48, *p* = 0.004) (Table 2). There was no significant correlation between changes in the sweat chloride test and Brody’s total score (r = 0.26, *p* = 0.274).

The analysis in child and adult groups showed a non-significant improvement in the total score and all subscores in the pediatric population. There was a significant improvement in the peribronchial thickening score in adults (before: 29.1 ± 10.3 vs. after: 26.9 ± 9.1; *p* = 0.044) and a non-significant improvement in the total score and mucous plugging.

BMI was significantly improved after treatment in both pediatric and adult populations (Table 3). There was a non-significant improvement in FEV1 in the total population as well as in the age subgroups (Table 1 and Table 3).

## 4. Discussion

To our best knowledge, this study is the first one to focus on CT changes induced by lumacaftor–ivacaftor in patients with cystic fibrosis. The main result is a significant improvement in the total CT score, especially related to an improvement in mucous plugging under combined therapy. 

These results are consistent with studies conducted with ivacaftor monotherapy in G551D-mutated patients, in which a significant improvement of peribronchial thickening was also demonstrated [11,12]. The effectiveness of ivacaftor monotherapy seems a little more important at short term follow-up (3–18 months after the start of ivacaftor) in the Chassagnon study with an improvement of 11% for total CT score and 32% for mucous plugging versus, respectively, 8% and 29% in our study with the lumacaftor–ivacaftor therapy. Sheikh et al. also found a significant improvement in bronchiectasis with ivacaftor monotherapy, whereas in the study by Chassagnon ed al. at long-term follow-up, bronchiectasis had an unfavorable course [11,12]. Our study found a non-significant improvement in bronchiectasis only in the pediatric population. 

The subgroup analysis also showed a significant improvement in peribronchial thickening in adults only. There was a non-significant improvement in total score and all subscores in the pediatric population and a non-significant improvement in total score and mucous plugging in the adult population. 

The discrete improvement in the subscores and the mostly non-significant results of the subgroup analysis could be explained on the one hand by a lower efficacy of the lumacaftor–ivacaftor combination compared to ivacaftor alone, and on the other hand probably by the lack of statistical power of our study.

However, six patients had a discrete improvement in bronchiectasis which may be responsible for a pseudo-increase of peribronchial thickening. In some patients, improvement in mucous plugging, sometimes major, may also be responsible for a pseudo-increase of bronchiectasis as already described by Chassagnon et al. [11]. These findings could minimize the real benefit of treatment on peribronchial thickening and bronchiectasis.

Although some components of the score were not significantly improved, there was also no significant deterioration in these subscores. Only a study versus placebo would really assess the impact of lumacaftor–ivacaftor on the CT scan changes, but such a study is no longer feasible from an ethical standpoint nowadays in France.

Chassagnon et al. found a good correlation between changes in FEV1 and total score, which is also the case in our study, where the total score as well as the mucous plugging and peribronchial thickening subscores appear to be a good reflection of FEV1 [11]. The good correlation between our CT scan scores and FEV1 supports the idea that CT scan is a reliable tool in the follow-up of cystic fibrosis patients, and potentially a useful tool for the evaluation of pulmonary anatomical lesions under treatment. No significant correlation was found between CT scan scores and the sweat chloride test but results of these tests were only available for 20 patients.

The subgroup analysis also found a significant improvement in BMI in both adults and adolescents, which is consistent with the study conducted under combined therapy in the real-life study by Burgel et al. [7]. Sweat chloride tests were available only for the pediatric population. The results were also significantly improved. By contrast, FEV1 improvement was present but not statistically significant in the total population (+0.6%) and in age subgroups (+0.3% in children, +1.3% in adults), unlike the cohort of Burgel et al. which included 845 patients for whom there was a significant improvement when treatment was taken continuously (+3.7%) [7]. This could probably be explained by a lack of power in our study, a non-dedicated design, and a higher mean FEV1 (75% versus 65%). CT scan changes are not specifically evaluated under therapy. 

Our study has several limitations: its retrospective design with sometimes missing data (especially expiring acquisitions) and acquisition parameters that may differ between the two centers involved. Delays between the baseline CT scan, the initiation of treatment and then the follow-up CT scan were highly variable, especially for the adult population, but it allowed us to expand our cohort.

The scans were read by only one radiologist, which did not allow inter-observer reproducibility, but this has already been assessed for Brody’s score [9,10]. Loeve et al. evaluated the intra- and inter-observer reproducibility of the Brody II scores, which were, respectively, 0.56 and 0.73 for peribronchial thickening, 0.79 and 0.65 for bronchiectasis, 0.77 and 0.79 for mucous plugging, and 0.77 and 0.80 for the total score [9]. It would have been preferable if the interpretation of the CT scans had been done anonymously, without knowing whether or not the patient was treated with dual therapy. However, simultaneous interpretation of baseline CT scan and follow-up CT scan limited intra-observer variability, by facilitating the detection of scanographic enhancement or degradation. Moreover, the radiologist was not aware of the clinical evolution of the patients at the time of interpretation.

Finally, our population including both adults and adolescents was heterogenous but it allowed us to obtain a larger study cohort.

Despite these limitations and the heterogeneity in the CT scan changes, the improvements in the total score and particularly in mucous plugging, which were sometimes dramatic in some patients, is a new argument in favor of the benefit provided by the lumacaftor–ivacaftor combination, whose effectiveness has been questioned due to the heterogeneity of the clinical response, its cost, and an improvement in FEV1 that was sometimes discreet [4,5,6].

Pulmonary involvement begins early in life, with about one third of patients with bronchiectasis within the first month of life [1]. Recently, the efficacy and safety of lumacaftor and ivacaftor have been demonstrated in the 2–5 and 6–11 years populations, allowing the indications to be extended to a broader pediatric population [13,14]. Otherwise recent data on the double tezacaftor– ivacaftor combination [15,16], and more recently, triple therapy, combining three CFTR modulators, elexacaftor–tezacaftor–ivacaftor, demonstrated extremely promising safety and clinical efficacy profiles [17,18,19,20,21].

The preservation of lung function is a clearly defined objective of CFTR modulating treatments. The evaluation of these revolutionary new therapies of CF, and their initiation at an earlier stage in life, are the main lines of research for the future. 

Improvements in artificial-intelligence programs could lead to a more standardized and reproducible approach in the follow-up of these patients, using low-dose CT scan or perhaps even more promising, with MRI.

## Figures and Tables

**Figure 1 jcm-10-01999-f001:**
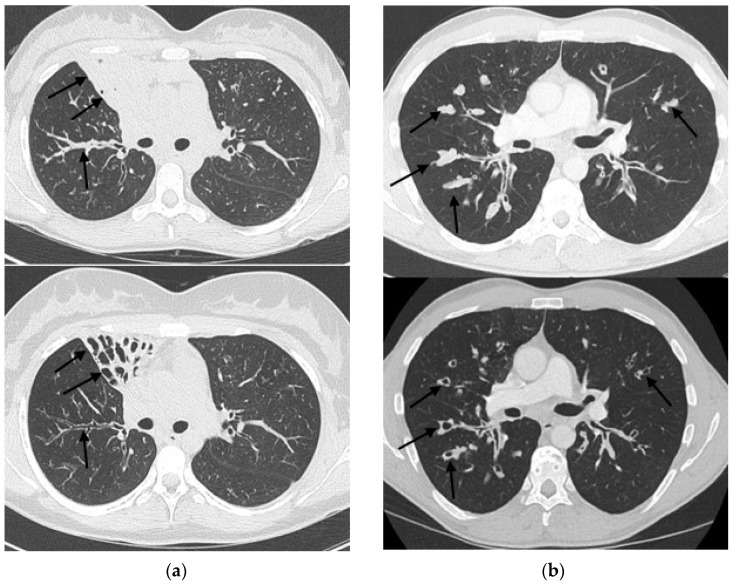
Two examples of CT scan before treatment (**top**) and after treatment (**bottom**) in two patients. (**a**) shows a complete regression of mucous pluggings (black arrows) in a collapsed middle lobe. (**b**) shows a complete regression of distal mucous pluggings (black arrows).

**Table 1 jcm-10-01999-t001:** Evolution before–after treatment of BMI, FEV1, sweat test and Brody score.

		Before Treatment	After/Under Treatment	*p*-Value
BMI (kg/m^2^)		19.2 ± 2.4	20.2 ± 2.8	<0.001
FEV1 (%)		74.8 ± 27.0	75.4 ± 25.4	0.737
Sweat chloride test (*n* = 20) (kg/m^2^)		112.7 ± 13.3	87.3 ± 17.1	<0.001
Brody’s score				
	Total score	65.5 ± 33.4	60.3 ± 29.6	0.049
	Bronchiectasis	23.3 ± 13.7	23.3 ± 14.5	0.991
	Mucous plugging	12.3 ± 8.8	8.7 ± 7.4	0.009
	Peribronchial thickening	23.8 ± 10.6	22.3 ± 9.1	0.241
	Parenchyma score	1.5 ± 1.7	1.5 ± 1.7	0.768
	Hyperinfilation score	6.7 ± 5.3	6.1 ± 5.1	0.373

**Table 2 jcm-10-01999-t002:** Correlations between changes in sweat test, FEV1, and Brody’s scores.

		Bronchiectasis	Mucous Plugging	Peribronchial Thickening	Parenchyma Score	Hyperinfilation Score	Total Score
Evolution of sweat chloride test (*n* = 20)	Pearson correlation coefficient (*p*-value)	0.12 (0.615)	0.16 (0.505)	0.21 (0.371)	0.25 (0.288)	0.28 (0.292) *	0.26 (0.274)
Evolution of FEV1 (*n* = 34)	Pearson correlation coefficient (*p*-value)	−0.06 (0.723)	−0.48 (0.004)	−0.43 (0.01)	−0.11 (0.547)	−0.26 (0.224) **	−0.51 (0.002)

* *n* = 16, ** *n* = 24.

**Table 3 jcm-10-01999-t003:** Evolution before–after treatment of BMI, FEV1, and sweat test in pediatric population.

	Before Treatment	After/Under Treatment	*p*-Value
BMI (kg/m^2^)	19.2 ± 2.4	20.2 ± 2.8	<0.001 ^a^
FEV1 (%)	74.8 ± 27.0	75.4 ± 25.4	0.737
Sweat chloride test (kg/m^2^)	112.6 ± 13.3	87.2 ± 17.1	<0.001

^a^ significant at *p* ≤ 0.05.

## Data Availability

The data presented in this study are available on request from the corresponding author.

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
