# Peer review of "Computed Tomographic Changes in Patients with Cystic Fibrosis Treated by Combination Therapy with Lumacaftor and Ivacaftor"

_jcm, 2021, doi:10.3390/jcm10091999_

Round 1
Reviewer 1 Report
The most commonly used surrogate endpoint in Cystic Fibrosis is spirometric variables such as the forced expiratory volume in 1 s (FEV1) and respiratory tract exacerbations. FEV1 is indirectly related to structural lung damage. There are drawbacks associated with it being insensitive to detect early and localised structural changes, is difficult for young children to perform and is not suitable for infants and most pre-school children. Computerised tomography (CT) images of the lungs offer a sensitive and accurate alternative to spirometry in CF patients. Although the study is interesting and presenting new information about HRCT changes in CF patients treated with CFTR modulators, some critical issues need to be addressed, as indicated in my specific comments.
1) Background in abstract needs to be revisited as it sounds more like an objective of the study than the background.
2) Be consistent with the nomenclature as it is phe508del in the abstract and F508del in the rest of the article.
3) Line 48-50 on page 2 of the introduction can be improved with more scientific language.
4) Is there any specific reason why sweat chloride was obtained only in the paediatric (adolescent) population?
5) Are there any threshold or consensus values set in the CF community or, more precisely, typical values that are used unanimously?
6) When authors say statistically significant does it mean it is clinically significant?
7) In figure 1, arrow indicators can be used to show bronchiectasis and bronchial wall thickening and mucus plugging as it would help the reader to see the pre and post CFTR modulator changes.
8) Also, it would be interesting if authors compare the values obtained in the current study vs ivacaftor monotherapy from previous studies.
9) Line 215 on page 6 spelling typo alter “CTsan” to “CT scan”.
Author Response
1) Background in abstract needs to be revisited as it sounds more like an objective of the study than the background.
Response (1) :Background has been reworded
2) Be consistent with the nomenclature as it is phe508del in the abstract and F508del in the rest of the article.
Response (2) The corrections has been made.
3) Line 48-50 on page 2 of the introduction can be improved with more scientific language.
Response (3) Line 48-50 on page 2 has been reworded
4) Is there any specific reason why sweat chloride was obtained only in the paediatric (adolescent) population?
Response (4) Data were missing because the sweat test is not routinely performed in the adult cystic fibrosis center (Hôpital Nord - Marseille). It is routinely performed only on the pediatric population in the pediatric CF center (Hôpital de la Timone - Marseille).
5) Are there any threshold or consensus values set in the CF community or, more precisely, typical values that are used unanimously?
Response (5) There is no established threshold or reference value for cystic fibrosis. There are multiple CT scores that are generally not used routinely. The modified Brody score is the most frequently used in the literature.
6) When authors say statistically significant does it mean it is clinically significant?
Response (6) The improvement seems to be statistically and clinically significant. Some patients show discrete clinical improvements, but others are dramatically improved, allowing them to be removed from the transplant lists. In our experience, no conventional therapy has produced such clinical or radiological improvement. The clinical impact was mainly evaluated by Burgel's and al study in 2020. Our clinical experience has allowed us to observe exceptional improvements for some patients. The Orkambi (c) therapy and other new specific therapies are a real revolution in the management of these patients. The clinical and radiological improvements seen in some patients are absolutely unprecedented.
7) In figure 1, arrow indicators can be used to show bronchiectasis and bronchial wall thickening and mucus plugging as it would help the reader to see the pre and post CFTR modulator changes.
Response (7) The arrows have been added
8) Also, it would be interesting if authors compare the values obtained in the current study vs ivacaftor monotherapy from previous studies.
Response (8) Data has been added from page 5-6, line 181-187
9) Line 215 on page 6 spelling typo alter “CTsan” to “CT scan”.
Response (9) The correction has been made
Reviewer 2 Report
Thank you for this interesting manuscript.
The topic is quite original.The paper is well written and the text is clear and easy to read.
The conclusions are consistent with the evidence and arguments presented, and they address the main question posed.
I only suggest rewriting the discussion in that order:
- short introduction;
- summary of main results without data and numbers;
- possible explanation of results obtained physiopathological term;
- comparison with previous studies;
- clinical implication of the results;
- limitations of the study;
- conclusions
Author Response
Thank you very much for your comments and suggestions.
Some data has been added in the comparison with previous studies (page 5-6, line 181-187).
Also, the discussion has been reorganized a bit. The paragraph on clinical implications has been positioned before the limitations of the study (page 6, line 211-213).